# The Genesis Mechanism and Health Risk Assessment of High Boron Water in the Zhaxikang Geothermal Area, South Tibet

**Liang Li** [1,2], **Yingchun Wang** [1,3,*], **Hongyu Gu** [4], **Lianghua Lu** [5], **Luping Li** [1,2], **Jumei Pang** [6] and **Feifei Chen** [7]

1    State Key Laboratory of Oil and Gas Reservoir Geology and Exploitation, Chengdu University of Technology, Chengdu 610059, China
2    Institute of Sedimentary Geology, Chengdu University of Technology, Chengdu 610059, China
3    College of Energy Resources, Chengdu University of Technology, Chengdu 610059, China
4    Chengdu Center, China Geological Survey, Chengdu 610081, China
5    Jiangsu Provincial Key Laboratory of Environmental Engineering, Jiangsu Provincial Academy of Environmental Science, Nanjing 210036, China
6    China Institute of Geo-Environment Monitoring, Beijing 100081, China
7    School of Engineering and Technology, China University of Geosciences, Xueyuan Road 29, Beijing 100083, China
*    Correspondence: wangyingchun19@cdut.edu.cn

**Abstract:** The natural discharge of geothermal water containing harmful components affects the water quality of the surrounding environment and brings security risks to drinking water safety. The geothermal water in Tibet is characterized by high boron content, but the water pollution caused by the discharge of this high boron geothermal water is not clear. In this study, we collected geothermal water and surface water from the Zhaxikang geothermal system in southern Tibet to investigate the causes of high boron geothermal water and the water pollution of water quality by its discharge. The results indicate that the hydrochemical type of geothermal water was $HCO_3$-Cl-Na, while that of cold spring water, mine water, river water, and lake water was $SO_4$-$HCO_3$-Ca-Mg. Hydrogen and oxygen isotopes show that the recharge source of cold groundwater was mainly snow-melting water and meteoric water, while in addition to that, there is magmatic water for hot springs. The boron content of geothermal water in the study area is as high as 42.36 mg/L, far exceeding the World Health Organization limit for drinking water (0.5 mg/L). The analysis of ion components and PHREEQC modeling indicated that the dissolution of silicate minerals and cation exchange controlled the composition of groundwater, and the boron in groundwater mainly came from the volatilization of magmatic components and the leaching of shallow sediments. The entropy weight water quality index was used to evaluate the water quality of the study area; about 42.9% of the groundwater samples are of good quality and can be used for drinking, mainly cold water that has not been mixed with geothermal water in the upstream. With the discharge of geothermal water into the river (with a mix ratio of ~20%), the downstream water quality gradually deteriorated. The health risk assessment of drinking water in the study area showed that the hazard index (HI) of drinking water in the mixed area was higher than 1 (with an average of 1.594 for children and 1.366 for adults), indicating that children are at a higher health risk than adults. Geothermal water with high boron content has been found all over the world, and the adverse effects of its natural drainage cannot be ignored.

**Keywords:** high boron water; evolution mechanism; water quality index; health risk assessment; Zhaxikang geothermal system

## 1. Introduction

The Qinghai–Tibet Plateau is well known as the "Asian Water Tower" (AWT) and is the source of the Yarlung Zangbo (Brahmaputra), Indus, Ganges, Mekong, Yangtze, and Yellow rivers. Its water environment is essential to the security of drinking water

for nearly 1.4 billion people in more than a dozen countries in Asia [1]. In addition, the Qinghai–Tibet Plateau is an important part of the Mediterranean–Himalayan Geothermal Belt, with lots of geothermal systems and numerous surface thermal manifestations [2–7]. Additionally, under the trend of energy transformation, geothermal energy as clean energy will undoubtedly usher in prosperity, but some potential water security risks cannot be ignored [8]. Compared with other types of natural water bodies (such as shallow groundwater and surface water), geothermal water with higher temperatures can usually leach more chemical components from its host medium. The typical harmful components include $CO_2$ and $H_2S$, as well as mercury, arsenic, fluorine, lead, chromium, boron, and radon [9]. Sources of these toxic elements are related to hydrochemical processes such as water–rock interaction, mixing, and degassing [10,11]. Under the influence of various geological or human factors, such as hot spring water discharge and geothermal water recharge, geothermal water is discharged into shallow aquifers, surface water, and the surrounding environment, bringing hidden dangers to the safety of surrounding drinking water [12,13]. Additionally, the rich geothermal resources in the Qinghai–Tibet Plateau will be accelerated exploitation driven by the long-term goal of carbon neutrality [14]. Therefore, the security of the water environment in the AWT and the assessment of potential health risks have become more significant as the anthropogenic accelerate the discharge of harmful components from geothermal water.

Geothermal fluids are usually a two-phase mixture with complex compositions, and the characteristic toxic and harmful substances carried by different geothermal fields are different [15]. Those toxic and harmful components will have a certain impact on the water quality around the drainage area. At present, the research on water quality in the Qinghai–Tibet Plateau mainly focuses on the effects and characteristics of geothermal water on surface water. Tian et al. [16] investigated the effects of arsenic, selenium, and fluoride on the quality of natural river water. Li et al. [17] found heavy metals such as arsenic, chromium, and lead in several rivers on the Qinghai–Tibet Plateau. Wang et al. [18] used the pollution evaluation model to study the heavy metal pollution of soil and groundwater in Tibet. However, the impact of high boron geothermal water in Tibet on the water environment has received little attention [19,20].

High boron groundwater has been reported worldwide, such as in Greece [21], Bangladesh and the US [22], Italy [23], Ghana [24], and China [25]. Boron concentrations in groundwater are generally low; however, geological processes such as geothermal activity [25], seawater intrusion into the aquifer [23], and human activities [26,27] would increase its concentration. The standard limit for boron in drinking water is 0.5 mg/L due to potential health risks [28]. High concentrations of B in drinking water can lead to low fertility and infertility, low fetal weight, and acute neurological and metabolic disorders [24]. Therefore, water safety risks associated with the discharge of high boron geothermal water cannot be ignored.

In China, high boron geothermal water is mainly distributed in the Yunnan–Tibet tropics [29]. In remote areas, residents often take shallow groundwater and river water as drinking water, while hot spring water is often discharged into river water without treatment. Therefore, we believe that the discharge of geothermal water with a high concentration of elements such as B and As will affect the surrounding water quality and bring hidden dangers to the drinking water safety of local residents.

The Zhaxikang geothermal system was chosen for this research. The main sources of drinking water in this area are river water and shallow groundwater, but the health risks of these water resources have not been reported. Based on the hydrochemical and isotopic characteristics of hot springs, cold springs, river water, cold groundwater, and lake water, the main aims of this study are (1) to identify and characterize the main components in hot springs, cold groundwater, and surface water; (2) to identify the source of major contaminants; (3) to use entropy weight water quality index (EWQI) to assess the water quality of the river and groundwater, and (4) to evaluate the potential health risks of boron

in geothermal water to local drinking water. The results are of great significance for the security and management of water resources in this region.

## 2. Materials and Methods

### 2.1. Study Area

The study area is located in the southern Tibetan and the northeastern part of the Himalayan mountains (E 91°55′–92°05′, N 28°15′–28°25′, Figure 1b). The Zhaxikang catchment area spans Longzi and Cuona counties in the Shannan district. Zhaxikang's terrain is high in the west and low in the east, with elevations ranging from 4300 m to 5600 m. This region is characterized by a plateau temperate semiarid monsoon climate.

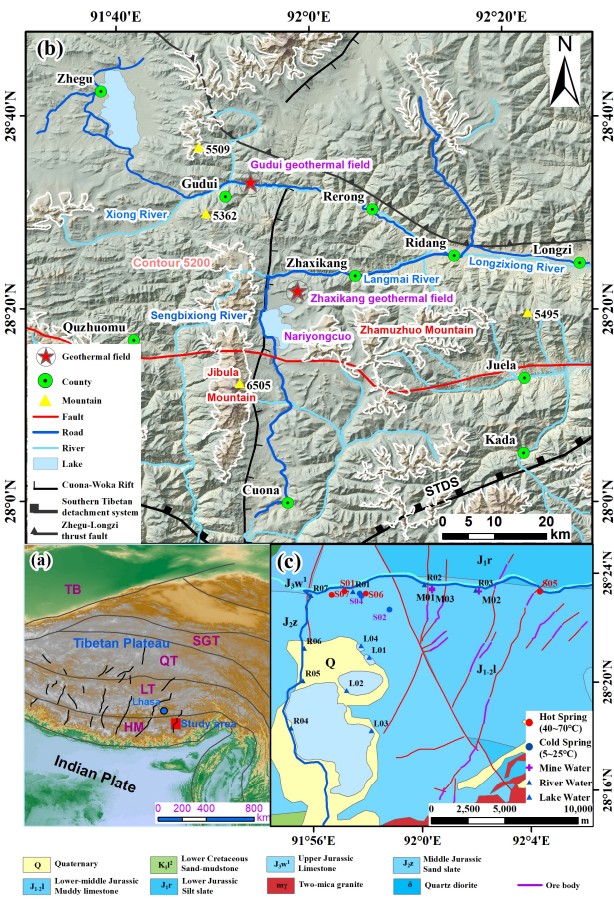

**Figure 1.** (**a**) Geographical distribution map of the study area; (**b**) tectonic and topography and (**c**) geology and sampling points distribution map of the study area. TB: Tarim Basin, SGT: Songpan–Ganzi Terrain, QT: Qiangtang Terrain, LT: Lasha Terrain, HM: Himalayan Mountains.

The study area is located at the intersection of the nearly east–west strike Zhegucuo–Longzi thrust belt and the nearly north–south extension Cuona–Woka rift, which is sandwiched by the Yarlung Zangbo Suture Zone and the southern Tibet Detachment System (Figure 1a) [30]. A series of near north–south trending extensional and torsional faults are arranged in an echelon in the study area (Figure 1c), and they are inferred to be the main channels for geothermal water to rise to the surface. Zhaxikang is an important metallogenic area in southern Tibet [31] and a large Sb-Pb-Zn-Ag deposit in the North Himalayas of southern Tibet [32]. Under the influence of artificial mining and geological processes, As and other metal elements will be released into the surrounding environment [33].

There are clusters of springs in an area of about 400 m² [34]. The temperature of the hot spring is from 43 °C to 70 °C, with a total flow rate of about 2 L/s. There is a slight

sulfur odor, and the hot water eventually flows into the Langmai River. Zhaxikang thermal area's reservoir temperature ranges between 185 °C and 205 °C [34].

### 2.2. Sampling and Analysis

In September 2018, water samples were collected in the study area. To analyze the spatial evolution of water quality in the Zhaxikang catchment, sampling points were placed at various potential groundwater and surface water interaction areas, as well as downstream of sewage discharge from mining areas that may cause pollution. For the mixing area between geothermal water and river water, sampling occurred upstream, in the mixing area, and downstream along the river. A total of 21 water samples were collected, including 11 surface water samples (4 from lakes in the southwest of the study area and the rest from various sections of the Langmai River), 4 hot spring water samples, 3 cold spring water samples, and 3 mine water samples (mine water at a depth of 200–400 m, representing cold groundwater). The Zhaxikang hot spring flows through the valley and then forms a thermal water pool that drains into the Langmai River. The distribution of sampling points for different types of water is shown in Figure 1c.

Each water sample was collected in a polyethylene bottle, which was rinsed three times with deionized water before sampling. The water sample for anion analysis was filtered through a 0.45 μm microporous membrane and stored in bottles. The sample used for cation analysis was also filtered and acidified with 6 N $HNO_3$ until pH < 2 and sealed for storage [35,36]. The water samples for hydrogen and oxygen isotope analysis were unfiltered and stored directly in polyethylene sampling bottles. In the field, a portable multiparameter (Hach HD40Q, Loveland, USA) was used to determine the temperature (°C), pH, total dissolved solids (TDS), oxidation-reduction potential (ORP), and electrical conductivity [37].

The water samples were tested and analyzed at the Institute of Geographic Sciences and Natural Resources, CAS (Chinese Academy of Sciences) within one month after the field works. Ion chromatography (ICS-1100, ThermoFisher Scientific, Waltham, USA) was used to measure anion concentrations, with an A SUPP5 150 anion separation column, conductance detector Suppressed CD (J002), and the eluents were 3.2 mmol/L $Na_2CO_3$ and 1.0mmol/L $NaHCO_3$, with a detection limit of 0.01 mg·$L^{-1}$. Inductively Coupled Plasma-Atomic Emission Spectrometry was used to determine the cations (ICP-AES, ThermoFisher Scientific, Waltham, USA). The cationic separation column was Metrosep C4-150/4.0, and the suppressed conductivity detector CD (J002), the leachate is 1.7 mmol/L $HNO_3$ and 0.7 mmol/L $C_7H_5NO_4$, and the detection limit was 0.01 mg·$L^{-1}$. Atomic fluorescence spectroscopy was used to measure $Fe^{3+}$, $Cu^{2+}$, $Pb^{2+}$, $Cr^{3+}$, $Ni^{2+}$, $Cd^{2+}$, and $As^{5+}$, with a detection limit of 0.06 mg/kg. Hydrogen and oxygen stable isotopes were determined using a laser water isotope analyzer (L1102-I, Picarro, Santa Clara, USA) and standard samples corresponding to the Vienna Standard Mean Ocean Water (VSMOW). The detection accuracy for $\delta^{18}O$ and $\delta D$ is ±0.1‰ and ±1‰, respectively. The ion charge balance error (%CBE) of each sample was calculated, and the results show that the %CBE of the cold groundwater and surface water samples is less than 5%, and that of hot spring water samples %CBE is less than 10%. This confirms the reliability of the ion analysis [38].

### 2.3. Data Processing and Analysis

This study uses the Entropy-weighted water quality index (EWQI) to describe the water quality characteristics of the study area. The calculation steps and results of EWQI are shown in Supplementary Tables S1 and S2 [39,40]. As a convenient and effective water quality assessment method, EWQI has been widely used by scholars all over the world (Supplementary Table S3) [39].

To assess the health risk of ground and surface water mixed with geothermal water, this study quantified the health risks from drinking water intake and dermal contact using the empirical model proposed by the United States Environmental Protection Agency [41],

and the adverse health effects were assessed at various ages (children and adults). The detailed calculation process and result can be found in Supplementary Table S4.

In this study, the B element was used as a tracer to calculate the mixing ratio of the river water near the upstream hot spring cluster. See Supplementary Table S4 for the calculation method. The activity of dissolved ions and the saturation index (SI) for various mineral phases were computed to analyze the equilibrium status of groundwater by using aqueous geochemical modeling software (PHREEQC, Version 3.6, Reston, USA), based on the WATEQ4F database [42]).

## 3. Results and Discussion

### 3.1. Hydrochemical Characteristics

Tables 1 and 2 show the statistical data of the physical and chemical parameters of 21 water samples in the Zhaxikang area and the standard acceptable limits for the drinking water standards of the WHO. It can be seen from Table 2 that the concentration of various ions in water exceeds the drinking water limits in the study area (Figure 2a,b).

**Table 1.** Physical parameters and hydrogen and oxygen isotopes of water samples from the Zhaxikang groundwater system.

| Sample ID | Water Type | pH | Temp [°C] | EC [µS/cm] | ORP [mV] | TDS [mg/L] | $\delta^2$H [‰] | $\delta^{18}$O [‰] |
|---|---|---|---|---|---|---|---|---|
| R01 | rw | 8.22 | 19.65 | 559 | 21.5 | 404.5 | −131 | −17.2 |
| R02 | rw | 8.79 | 17.84 | 604.5 | 20 | 455.5 | −133 | −17.1 |
| R03 | rw | 8.73 | 16.94 | 613 | 31.8 | 470.9 | −141 | −18.2 |
| R04 | rw | 8.2 | 5.55 | 343 | 27.8 | 354.8 | −129 | −17.8 |
| R05 | rw | 8.19 | 6.33 | 302.9 | 35 | 305.9 | −128 | −17.6 |
| R06 | rw | 8.68 | 15.08 | 232.4 | 25.6 | 186.3 | −128 | −17.4 |
| R07 | rw | 8.8 | 17.17 | 284.6 | 12.6 | 217.6 | −129 | −17.2 |
| L01 | lw | 8.94 | 18.34 | 281.1 | 23.2 | 209.3 | −128 | −17.4 |
| L02 | lw | 9.24 | 14.89 | 8944.7 | 23.2 | 7206 | −76.7 | −8.26 |
| L03 | lw | 8.74 | 10.66 | 2630.7 | 25.4 | 2356 | −67 | −6.59 |
| L04 | lw | 9.65 | 16.1 | 422.6 | 29.8 | 331.2 | −99.3 | −11.4 |
| M01 | mw | 8.48 | 19.2 | 793.2 | 40.3 | 580.1 | −157 | −20.1 |
| M02 | mw | 8.05 | 15.65 | 935.6 | −41 | 740.4 | −154 | −21.1 |
| M03 | mw | 7.74 | 21.4 | 718.4 | 38.7 | 554 | −156 | −20.3 |
| S01 | hsw | 6.82 | 50.2 | 3523.1 | −10.6 | 1547 | −144 | −17.3 |
| S02 | csw | 8.32 | 6.53 | 240.5 | 23.3 | 241.5 | −147 | −19 |
| S03 | csw | 6.75 | 8.17 | 2035.4 | 15.6 | 1950 | −144 | −17 |
| S04 | csw | 8.6 | 18.35 | 1152 | 14.5 | 857.7 | −109 | −11.5 |
| S05 | hsw | 7.03 | 49.97 | 2372.1 | −19.6 | 1044 | −147 | −19.5 |
| S06 | hsw | 7.01 | 43.33 | 4138.7 | −17.7 | 1992 | −142 | −16.3 |
| S07 | hsw | 7.35 | 69.51 | 4698 | −262.5 | 1653 | −142 | −16.6 |
| Snow-melt water [a] | | - | - | - | - | - | −175.0 | −24.5 |
| Magmatic water [b] | | - | - | - | - | - | −20 ± 10 | 10 ± 2 |

Note: rw: river water, lw: lake water, mw: mine well water, csw: cold spring water, hsw: hot spring water; "-" indicates no value; [a], [43]; [b], [44].

**Table 2.** Statistical results of chemical parameters for different water types in the study area (unit: mg/L).

| Parameters | WHO Standards | Hot Spring Water (n = 4) | | | Cold Spring Water (n = 3) | | | Lake Water (n = 4) | | | Mine Water (n = 3) | | | River Water (n = 7) | | |
|---|---|---|---|---|---|---|---|---|---|---|---|---|---|---|---|---|
| | | Min | Max | Mean | Min | Max | Mean | Min | Max | Mean | Min | Max | Mean | Min | Max | Mean |
| $K^+$ | ≤200 | 11.68 | 35.40 | 26.31 | 0.54 | 26.10 | 12.75 | 0.54 | 54.23 | 17.65 | 0.35 | 2.25 | 1.45 | 0.42 | 7.80 | 3.02 |
| $Na^+$ | ≤200 | 193.00 | 695.00 | 486.25 | 8.43 | 584.00 | 288.14 | 5.73 | 1028.00 | 318.15 | 8.75 | 28.30 | 20.25 | 4.58 | 69.50 | 27.87 |
| $Ca^{2+}$ | ≤200 | 44.20 | 151.90 | 107.30 | 58.61 | 150.90 | 90.66 | 7.51 | 76.31 | 35.76 | 83.04 | 149.30 | 106.39 | 43.89 | 92.58 | 61.60 |
| $Mg^{2+}$ | ≤150 | 7.43 | 28.07 | 17.77 | 18.81 | 47.85 | 32.91 | 13.08 | 2123.00 | 671.71 | 59.71 | 84.46 | 70.19 | 12.78 | 43.63 | 19.54 |
| $Cl^-$ | ≤250 | 128.00 | 621.00 | 433.55 | 7.60 | 530.80 | 258.83 | 0.11 | 103.50 | 31.35 | 1.64 | 10.05 | 5.07 | 0.00 | 69.83 | 22.86 |
| $SO_4^{2-}$ | ≤250 | 175.30 | 235.90 | 201.00 | 66.74 | 163.20 | 104.09 | 69.61 | 8524.00 | 2654.55 | 161.40 | 427.20 | 255.13 | 72.15 | 201.40 | 128.77 |
| $HCO_3^-$ | ≤120 | 622.62 | 1401.00 | 1004.00 | 186.79 | 1556.54 | 840.53 | 62.26 | 1867.85 | 786.05 | 311.31 | 466.96 | 394.32 | 93.39 | 280.18 | 191.23 |
| $NO_3^-$ | ≤20 | 0.00 | 5.82 | 1.55 | 0.19 | 23.98 | 9.93 | 0.00 | 0.70 | 0.23 | 0.00 | 1.04 | 0.45 | 0.00 | 1.72 | 0.49 |
| B | <0.5 | 8.55 | 66.36 | 42.36 | 0.35 | 57.02 | 30.02 | 0.02 | 0.24 | 0.08 | 0.07 | 0.99 | 0.66 | 0.01 | 9.46 | 3.52 |
| $SiO_2$ | - | 54.54 | 105.50 | 72.23 | 7.96 | 24.82 | 13.96 | 1.51 | 5.88 | 3.12 | 13.09 | 25.64 | 19.87 | 4.47 | 20.44 | 12.22 |
| V (µg/L) | - | 1.55 | 4.04 | 3.23 | 0.62 | 4.45 | 3.35 | 0.95 | 2.52 | 1.71 | 0.28 | 0.43 | 0.35 | 0.38 | 1.46 | 0.93 |

**Table 2.** *Cont.*

| Parameters | WHO Standards | Hot Spring Water (n = 4) | | | Cold Spring Water (n = 3) | | | Lake Water (n = 4) | | | Mine Water (n = 3) | | | River Water (n = 7) | | |
|---|---|---|---|---|---|---|---|---|---|---|---|---|---|---|---|---|
| | | Min | Max | Mean | Min | Max | Mean | Min | Max | Mean | Min | Max | Mean | Min | Max | Mean |
| As (μg/L) | ≤50 | 0.03 | 6.96 | 2.86 | 0.03 | 5.92 | 1.99 | 0.03 | 126.68 | 34.56 | 0.03 | 4.50 | 2.98 | 0.03 | 2.22 | 0.34 |
| Cd (μg/L) | <5 | 0.36 | 0.46 | 0.41 | 0.45 | 0.39 | 0.42 | 0.42 | 0.54 | 0.46 | 0.48 | 0.57 | 0.53 | 0.35 | 0.58 | 0.42 |
| Pb (μg/L) | <10 | 0.64 | 0.70 | 0.66 | 0.62 | 0.65 | 0.63 | 0.59 | 0.69 | 0.71 | 0.65 | 0.68 | 0.66 | 0.64 | 0.66 | 0.65 |
| Fe (μg/L) | <300 | 28.60 | 108.89 | 48.37 | 25.04 | 65.88 | 47.52 | 3.85 | 163.09 | 57.06 | 43.56 | 103.82 | 66.03 | 31.66 | 72.31 | 51.15 |
| pH | 6.5~8.5 | 6.82 | 7.35 | 7.05 * | 6.75 | 8.60 | 7.89 * | 8.74 | 9.65 | 9.14 * | 7.74 | 8.48 | 8.09 * | 8.19 | 8.80 | 8.52 * |
| TDS | ≤1000 | 1044.00 | 1992.00 | 1559.00 | 241.50 | 1950.00 | 1016.40 | 209.30 | 7206.00 | 2525.63 | 554.00 | 740.40 | 624.83 | 186.30 | 470.90 | 342.21 |

Note: * The pH is just a statistical mean value.

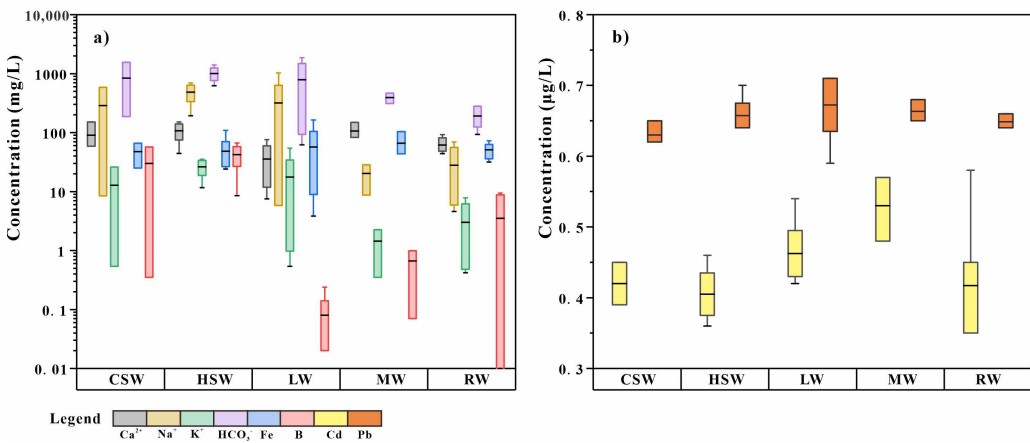

**Figure 2.** Box plots of chemical parameters for different water types in the study area.

The spring water is characterized by high concentrations of boron and bicarbonate (Figure 2a). This is a common feature of Tibetan hot springs [29]. The ORP value of hot spring water is negative, which is lower than that of other types of water, indicating reductive conditions for the deep hot water. Compared with cold spring water, hot spring water can leach out more surrounding rock components and has a higher ion concentration.

The positive ORP value of the lake water indicates an oxidizing environment. Additionally, the average temperature of the lake is 15 °C. Compared with other types of surface water, lake water has weaker discharge conditions and stronger evaporation.

The average pH value of the mine well water is 8.09, slightly alkaline. Sample M02, with an ORP value of −41.0 mV, is located downstream of the mining area, indicating a decrease in groundwater flow rate. The higher Fe concentration of M02 may be due to the less precipitation process under reduction conditions.

The variation in the hydrochemical parameters of the river water samples is large but regular, indicating the mixing of river water from other sources. Along the flowing direction from R04 to R05 (Figure 1c), the water temperature gradually increases, and the concentration of B also increases significantly. This is consistent with the results of the influence of groundwater containing arsenic, fluorine, and boron on drinking water in Argentina studied by Hudson-Edwards et al. [45].

### 3.2. Evolution Mechanism of Hydrochemical Components

#### 3.2.1. Source of Geothermal Water

The relationship between hydrogen and oxygen isotopes in groundwater can be used to infer its origin [34,46]. The relationship between $\delta^{18}O$ and $\delta D$ in the study area is shown in Figure 3.

Surface water and groundwater are commonly distributed near the Global Meteoric Water Line (GMWL), with meteoric water as the recharge source [47]. The isotope data representing ice–snow meltwater collected from the Nyenqing–Tanggula Mountain at an altitude of approximately 5500 m are also shown in Figure 3. [43], and magmatic water with

$\delta$D of 20‰ $\pm$ 10‰ and $\delta^{18}$O of 10‰ $\pm$ 2‰ from Giggenbach [44]. All of the hot spring samples were distributed within the mixing area of ice–snow melting water and magma water, indicating that they were recharge sources of geothermal water. The $\delta^{18}$O and $\delta$D values of S05 and S02 are lower and closer to GMWL, and the other spring water samples have higher values, indicating that S05 and S02 are supplied by a greater proportion of meteoric water, whereas other springs have experienced greater deep source mixing during migration [48]. According to Figure 3, ice melting and local meteoric waters are the recharge source of river water and mine water. The high $\delta^{18}$O and $\delta$D values of water collected from the lake, especially L02 and L03, indicate that the lake water occurred with strong evaporation and formed an evaporation line with $R^2$ = 0.9913.

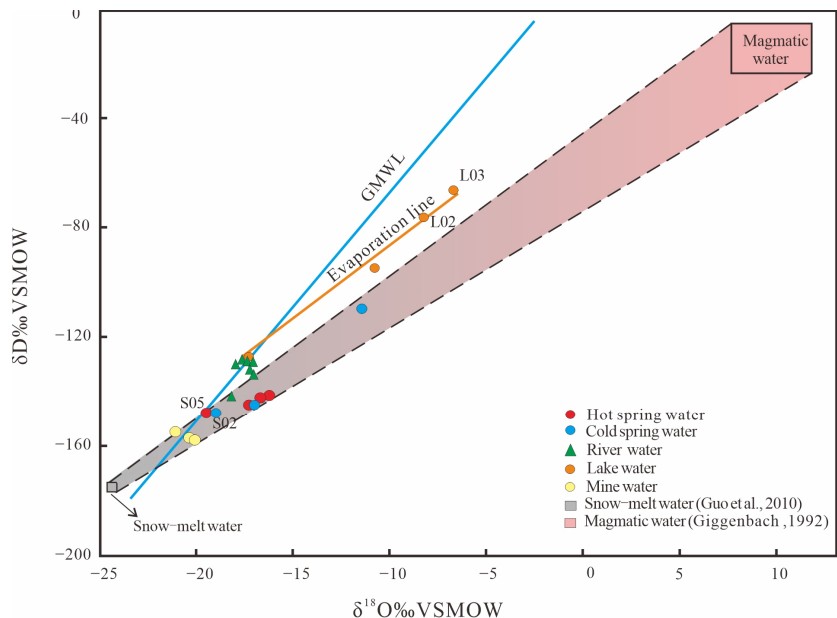

**Figure 3.** Relationship of $\delta$D vs. $\delta^{18}$O in surface water, mine water, cold spring, and hot spring [43,44].

### 3.2.2. Hydrochemical Types and Water–Rock Interaction

According to the piper ternary diagram (Figure 3), the hot spring water in the study area is located in zone III of the diagram and has the hydrochemical type $HCO_3$-Cl-Na. Hydrochemical types of mine water and river water are $SO_4$-$HCO_3$-Ca-Mg (Figure 4).

The hot spring water samples fall in region D of the cationic triangle on the left, indicating that the cations are dominated by $Na^+$. The lake and river water samples fall in zones A and B, suggesting that the cations in the surface water are mainly nondominant and calcium types. It can be seen that $HCO_3^-$ is the main anion, and the lake water was dominated by sulfuric acid type, which could be affected by evaporation.

The ion concentration in groundwater is determined by the hydrogeochemical processes that occur in the aquifer system [8,49]. Gibbs [50] classified the natural factors that control the chemical characteristics of groundwater as precipitation, evaporation, and water–rock interactions. The Gibbs diagram of the Zhaxikang area is shown in Figure 4.

In the study area, except for a few points that fall outside the anion ratio, almost all samples are controlled by water–rock action within the Gibbs map (Figure 5a). In the Gibbs cation ratio (Figure 5b), most samples are controlled by water–rock interaction, and some hot spring samples fall near the region controlled by evaporation. As a result, the chemical evolution of groundwater is primarily controlled by water–rock interaction and partly by evaporation. Furthermore, the increase in the $Na^+/(Na^+ + Ca^{2+})$ ratio with increasing TDS indicates that the groundwater may be altered by cation exchange between $Na^+$ and $Ca^{2+}$ [24].

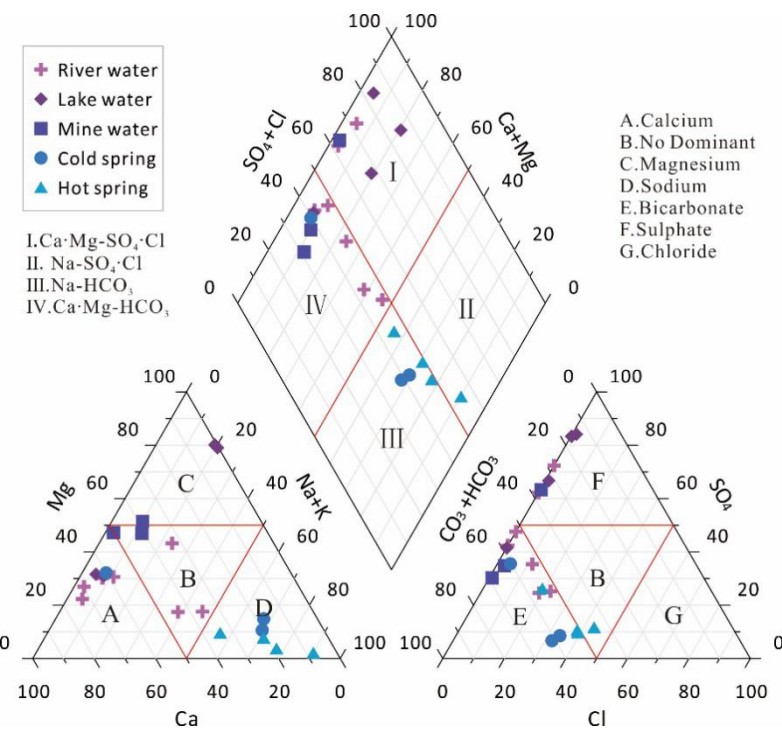

**Figure 4.** Piper diagram of river water, lake water, spring water and mine water.

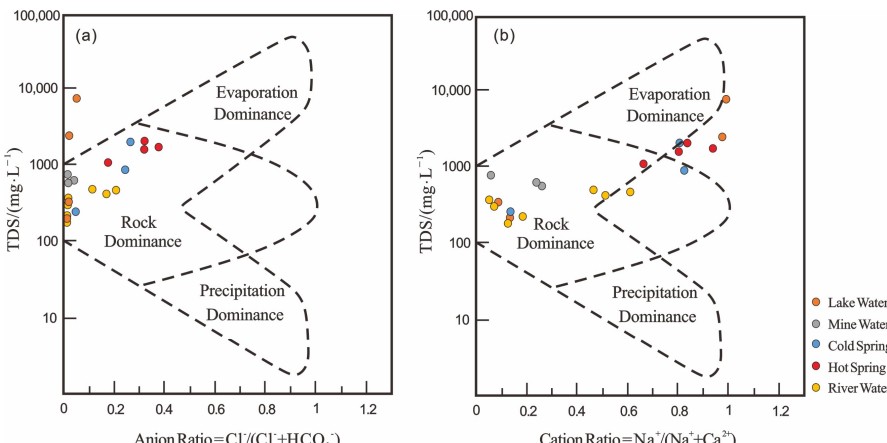

**Figure 5.** Gibbs diagrams of groundwater samples. (**a**) $\rho Na^+ / \rho(Na^+ + Ca^{2+})$; (**b**) $\rho Cl^- / \rho(Cl^- + HCO_3^-)$.

Through the above analysis, the types of water in the study area are $HCO_3$-Cl-Na and $SO_4$-$HCO_3$-Ca-Mg. Their water chemical composition is controlled by water–rock interaction and evaporation, and it is necessary to further analyze the water–rock reaction process they experience.

3.2.3. Mineral Saturation States and Partial Pressure of $CO_2$ (g)

Figure 6 shows the mineral saturation index (SI) of different types of groundwater and surface water in the study area along the flow direction. In general, the evaporite minerals (gypsum, anhydrite, and halite) are all undersaturated (SI < 0), indicating that the water in Tibet can still continue dissolving evaporite minerals. The climate in Tibet is dry, with strong weathering capacity and evaporation. Therefore, it is possible to provide more sources of evaporite minerals to river water [51].

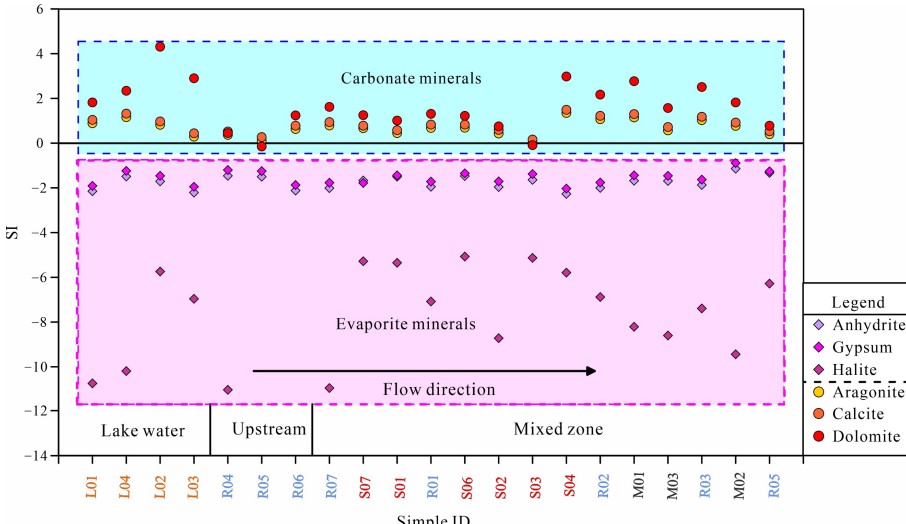

**Figure 6.** Saturation index (SI) of potentially relevant minerals for the water samples in the study area.

For carbonate minerals, the analysis results of all water samples show a supersaturated state (SI > 0) and show a certain regularity. Along the flow direction of the river water, different types of groundwater discharge into the river water. The SI value of river water relative to carbonate minerals gradually increased, indicating that its saturation state gradually increased, which also shows that the impact of groundwater discharge cannot be ignored.

Figure 7 shows the relationship between $HCO_3^-$ + $CO_3^{2-}$ and the logarithm of partial pressure of $CO_2$ (g) (log (p$CO_2$ (g))) for the groundwater samples. Most surface water and groundwater fall within the soil (p$CO_2$ (g) = $10^{-1.5}$ atm) and atmospheric (p$CO_2$ (g) = $10^{-3.5}$ atm) $CO_2$ (g) range. Surface water is very close to equilibrium with atmospheric $CO_2$ (g), indicating the fastest recharging waters [52]. The hot spring samples are in equilibrium with values above the upper limit of $CO_2$ (g) in the soil region (p$CO_2$ (g) = $10^{-1.5}$ atm), indicating that they are influenced by volcanic emissions of $CO_2$ (g). Their p$CO_2$ (g) values ranged from $10^{-1.13}$ to $10^{-0.66}$ atm.

The $CO_2$ in the open system will be continuously replenished, and the p$CO_2$ of the groundwater in the study area is greater than the atmospheric value, indicating that the groundwater system is an open system. Additionally, the study area is located in the recharge–recharge runoff area, with strong water alternation, which is conducive to the migration of ions and the continuous replenishment of atmospheric $CO_2$. p$CO_2$ also controls the concentration of $Ca^{2+}$ and $HCO_3^-$ in water by controlling the solubility of carbonate minerals.

### 3.2.4. Source of Chemical Components

Because of water–rock interactions, silicate minerals such as plagioclase and albite are common sources of Na and K in groundwater [53]. The process of $Na^+$ replacing $Ca^{2+}$ and $Mg^{2+}$ in the cation exchange process may also be a major factor in cation change [54]. The interrelationships between major elements can be used to identify the water–rock processes that groundwater encounters [55].

The $Na^+$-$Cl^-$ diagram that cold spring, mine water, and surface water are all located under the curve of y = x (Figure 8a), and halite dissolution can be excluded. The Cl in groundwater generally comes from weathering, the leaching of rocks or sediments, or volatilization from deep magma. The hot springs have offset the curve of y = x (Figure 8a), indicating volatiles from deep magma [55]. The extra $Na^+$ in cold water is most likely the result of silicate weathering and/or cation exchange processes.

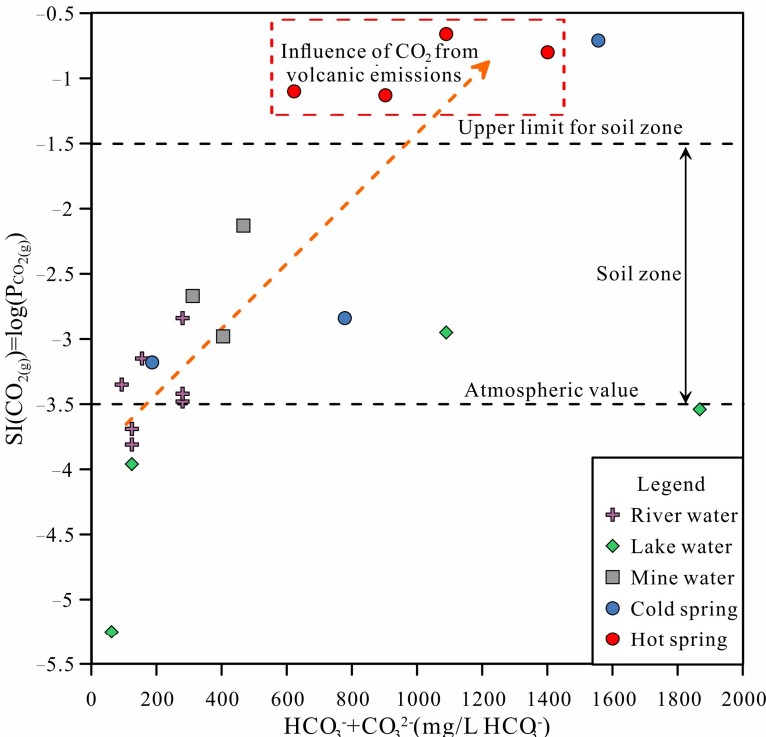

**Figure 7.** Relationship between $HCO_3^-$ + $CO_3^{2-}$ and log ($pCO_2$ (g)) for the water samples in the study area.

If the relationship between $Ca^{2+}$ and $SO_4^{2-}$ in water is on the y = x curve (Figure 8b), it indicates that they are primarily derived from gypsum dissolution. However, the groundwater samples in the study area are basically below the curve of y = x, which is inconsistent with the dissolution mechanism of gypsum, and cation exchange seems to be a better explanation for this phenomenon. In addition, the data of groundwater samples fall on the three regions divided by the curves of y = x and y = 2x from the relationship of $Ca^{2+}$ and $HCO_3^-$ (Figure 8c). Hot springs and some cold springs are above y = 2x, indicating that $HCO_3^-$ comes from somewhere other than dissolution, such as deep metamorphic degassing or magma volatilization. Those between the two curves are thought to be the result of calcite dissolution.

Furthermore, the main rock types involved in the water–rock reaction can be identified using the $Mg^{2+}/Na^+$ vs. $Ca^{2+}/Na^+$ and $HCO_3^-/Na$ vs. $Ca^{2+}/Na^+$ (Figure 8d,e) [56]. According to Figure 8d,e, mine water is located in the transition zone of silicate and carbonate dissolution, while hot springs and some cold springs are located in the transition zone of silicate and evaporite. In Figure 8f, most of the samples are located above the isocline, which indicates that the $Mg^{2+}$ and $Ca^{2+}$ in groundwater are mainly from aluminosilicate dissolution, and carbonate dissolution may also exist. Excess $Na^+$ in groundwater could be caused by silicate weathering. According to Figure 8g, the geothermal water samples mostly fall in the upper part of the region, which is affected by carbonate decomposition, while some samples of surface water and cold spring water fall in the lower part, indicating that they are also affected by evaporative decomposition. In addition, diagram $Na^+ + K^+ - Cl^-$ vs. $Ca^{2+} + Mg^{2+} - (HCO_3^- + SO_4^{2-})$ can also indicate the degree of cation exchange Figure 8g.

The chemical composition of Zhaxikang water was analyzed by PHREEQC, and Figure 9A–C shows the projections of all water samples in the stability diagrams for Na-, K-, and Ca-feldspars. It can be used to determine the feldspar weathering stage and the corresponding secondary minerals in the study area.

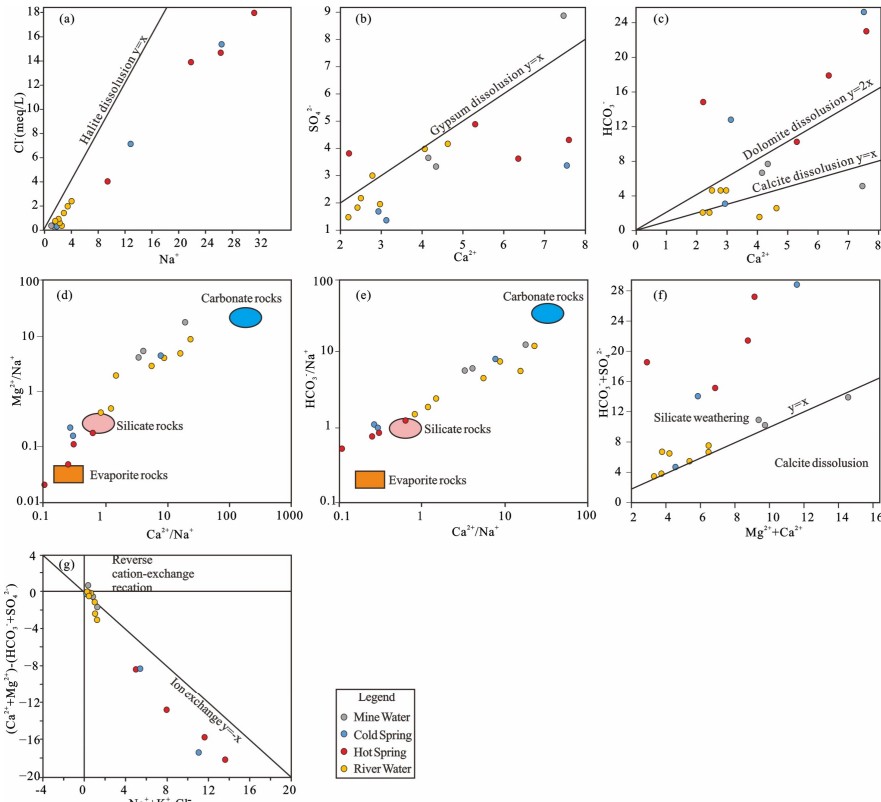

**Figure 8.** Relationship diagrams of (**a**) $Cl^-$ vs. $Na^+$, (**b**) $SO_4^{2-}$ vs. $Ca^{2+}$, (**c**) $HCO_3^-$ vs. $Ca^{2+}$, (**d**) $(Mg^{2+}/Na^+)$ vs. $(Ca^{2+}/Na^+)$, (**e**) $(HCO_3^-/Na^+)$ vs. $(Ca^{2+}/Na^+)$, (**f**) $HCO_3^- + SO_4^{2-}$ vs. $Ca^{2+} + Mg^{2+}$, (**g**) $Na^+ + K^+ - Cl^-$ vs. $Ca^{2+} + Mg^{2+} - (HCO_3^- + SO_4^{2-})$.

It can be seen from Figure 9 that the water samples in the study area are in a state of dissolution with respect to feldspar. The weathering process of albite in groundwater and surface water has passed from the gibbsite stage to the kaolinite stage, and the hot spring water samples have reached the stage of Na-montmorillonite (Figure 9A). The K-feldspar has not yet reached saturation and is in a state of dissolution, and the weathering has reached the kaolinite stage (Figure 9B). Weathering of anorthite is also at the kaolinite stage (Figure 9C).

The groundwater in the study area is supersaturated relative to quartz and undersaturated relative to amorphous silica (Figure 9). Under the undersaturated state, the solubility of amorphous silica controls the upper limit of $H_4SiO_4$ activity in the water sample. In shallow water areas, $CO_2$ gas participates in the hydrolysis of silicate minerals to form gibbsite, which in turn forms kaolinite, then up to the montmorillonite stage and produces amorphous silica. The weathering of feldspar and its secondary minerals requires the consumption of $H_2CO_3$. While $H_2CO_3$ is related to $pCO_2$, the content of $Ca^{2+}$ and $HCO_3^-$ in water is proportional to $Pco_2$, indicating that $Ca^{2+}$ and $HCO_3^-$ are formed by the dissolution of carbonate and feldspar minerals by $CO_2$, which makes the two ions accumulate continuously.

In conclusion, the ions in the water samples are mainly controlled by cation exchange and silicate weathering, and the related ions in hot spring water samples are mainly from sodium montmorillonite. The non-aluminosilicate minerals that control the dissolution and balance of chemical composition in water are mainly calcite, and the aluminosilicate minerals that participate in the water–rock reaction are mainly feldspar, followed by kaolinite and montmorillonite.

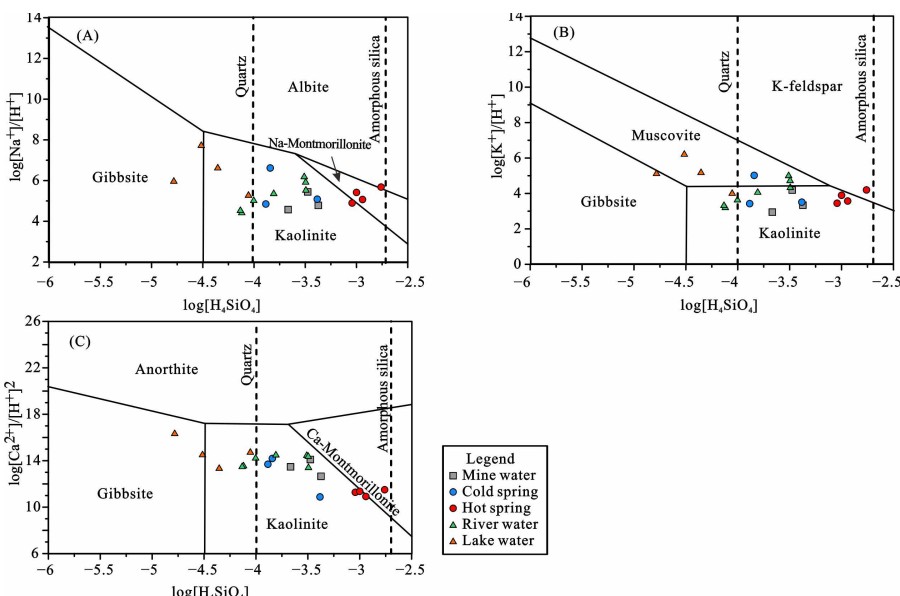

**Figure 9.** (**A–C**): Stability diagrams of Na-, K-, and Ca-feldspars and their weathering products for water samples from the study area. The dashed lines represent saturation with respect to quartz and amorphous silica.

### 3.2.5. Source of B in The Groundwater

Boron is the main component of magmatic rock or fine-grained soil and often forms borates in water environments [57], and Its mobility is closely related to the pH of the fluid [58]. B concentration in geothermal water in the study area is relatively high, ranging from 32 to 66 mg/L, with an average value of 43 mg/L. When they are discharged into a river, the concentration of B downstream can be as high as 9 mg/L, exceeding the 0.5 mg/L drinking water standard. According to the analysis results of PHREEQC, B mainly exists in the geothermal water in the form of $H_3BO_3$ and $H_2BO_3{}^-$. The geothermal water environment in the Zhaxikang area is characterized by high temperature, high TDS, and strong reducibility, which is conducive to boron enrichment (Supplementary Figure S1) [59].

Pearson correlation analysis was performed on 13 hydrochemical parameters in Zhaxikang to explain the influencing factors of groundwater quality, and the value of the phase relationship is shown in Table 3 [60]. B is well correlated with most ions, and $Cl^-$ (r = 0.99) and $HCO_3{}^-$ (r = 0.63) are well correlated, which may be due to the dissolution of carbonate minerals [59]. pH is negatively correlated with bicarbonate and B, indicating that an acidic environment enhances bicarbonate and B enrichment.

B can be parsed into groundwater by flushing clay minerals with freshwater [22]. However, cation exchange is also accompanied by the process of freshwater flushing, resulting in Ca depletion and Na enrichment in groundwater, which is consistent with the results of the study area. Furthermore, B may be derived from deep magma volatilization in the geothermal area [25]. A portion of B enters the river directly from hot spring discharge, while another portion is adsorbed on the surface of clay or muddy minerals in near-surface sediments. It is clear from the good relationship between B, Na, and Cl that B is controlled by adsorption/dissolution in groundwater.

### 3.3. Relationship between the Surface Water and Groundwater

The study area is dominated by inorganic pollutants from geothermal water. The mixing of cold groundwater and deep hot water is the main reason for the differences in the properties and composition of hot springs. Surface water and groundwater from a complete water cycle are controlled by the geological structure and other factors.

**Table 3.** Correlation coefficients of components for the collected sample.

| | pH | TDS | K$^+$ | Na$^+$ | Ca$^{2+}$ | Mg$^{2+}$ | Cl$^-$ | SO$_4^{2-}$ | HCO$_3^-$ | NO$_3^-$ | As | Cd | B |
|---|---|---|---|---|---|---|---|---|---|---|---|---|---|
| pH | 1.00 | | | | | | | | | | | | |
| TDS | 0.01 | 1.00 | | | | | | | | | | | |
| K$^+$ | −0.33 | 0.88 | 1.00 | | | | | | | | | | |
| Na$^+$ | −0.35 | 0.86 | 0.99 | 1.00 | | | | | | | | | |
| Ca$^{2+}$ | −0.71 | −0.18 | 0.04 | 0.10 | 1.00 | | | | | | | | |
| Mg$^{2+}$ | 0.32 | 0.93 | 0.65 | 0.63 | −0.39 | 1.00 | | | | | | | |
| Cl$^-$ | −0.73 | 0.29 | 0.70 | 0.73 | 0.49 | −0.07 | 1.00 | | | | | | |
| SO$_4^{2-}$ | 0.31 | 0.93 | 0.66 | 0.64 | −0.37 | 0.99 | −0.05 | 1.00 | | | | | |
| HCO$_3^-$ | −0.42 | 0.84 | 0.93 | 0.95 | 0.16 | 0.59 | 0.70 | 0.59 | 1.00 | | | | |
| NO$_3^-$ | −0.46 | 0.09 | 0.22 | 0.28 | 0.44 | −0.09 | 0.47 | −0.10 | 0.40 | 1.00 | | | |
| As | 0.30 | 0.92 | 0.67 | 0.65 | −0.38 | 0.99 | −0.03 | 0.99 | 0.58 | −0.11 | 1.00 | | |
| Cd | 0.16 | 0.27 | 0.06 | 0.03 | 0.04 | 0.35 | −0.25 | 0.34 | 0.14 | 0.04 | 0.34 | 1.00 | |
| B | −0.72 | 0.18 | 0.61 | 0.64 | 0.51 | −0.18 | 0.99 | −0.17 | 0.63 | 0.48 * | −0.14 | −0.27 | 1.00 |

Note: * represents the statistical mean value.

River water can reduce the concentration of harmful elements through physical precipitation and ion exchange. The changes of Cd, Pb, B, V, and Fe along the river direction are shown in Figure 10a. B and V will increase significantly in the downstream river as the flowing water passes through the discharge area of the hot spring, resulting in a change in the river water quality of the river. According to the calculation of the mixing ratio of surface water B, the mixing ratio of geothermal water to river water is about 20.3%. Although the river water has a certain self-purification ability, the ratio downstream is still 14.3%. The concentration of B (B = 9.46 mg/L) still exceeded the standard drinking water limit (Figure 10b).

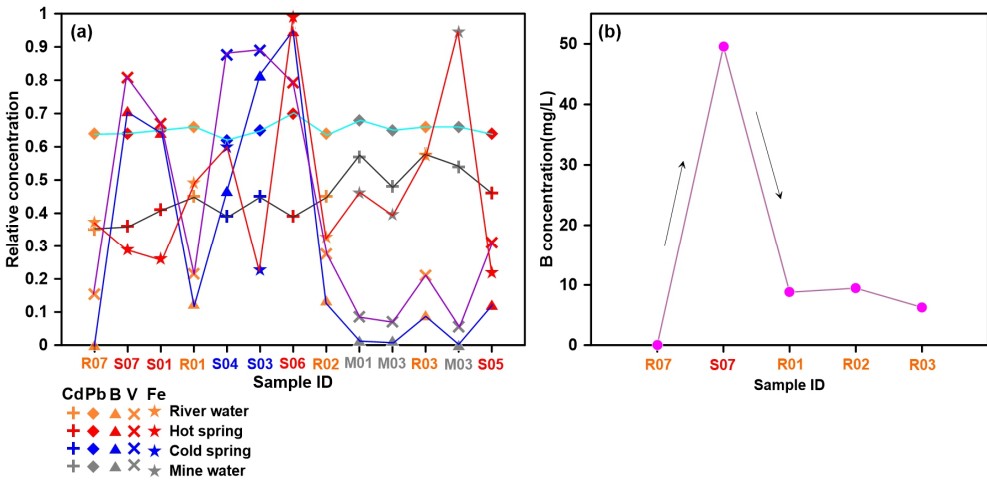

**Figure 10.** (**a**) The covariant relationship of Cd, Pb, B, V and Fe along the river direction; (**b**) Variations of B concentrations of Langmai River waters along the flow direction.

The elements such as B and As carried by geothermal water are significantly higher than those carried by surface water. In most geothermal areas of Tibet, such as Yangbajing, Yangyi, Tasha, Quzhuomu, Semi, and Mansarovar, water from geothermal wells or springs is discharged into rivers nearby directly. As a result, geothermal water will certainly affect the quality of the surrounding surface water. Furthermore, as hot springs rise from the subsurface to the surface, many toxic and harmful elements accumulate in the shallow sediments, and they are released into nearby rivers and lakes through processes such as rainfall, weathering, and mineral alteration, contaminating surface water quality on a long-term scale.

### 3.4. Health Risk Assessment

Boron is an essential trace element in plants, but it can cause damage to the human liver and kidney in high concentrations [61]. Human immune function may also be compromised if excessive amounts of boron are consumed in drinking water over a long period [62]. Most of the rural areas in Tibet are built along rivers, and the polluted water will inevitably affect the drinking water of residents in Zhaxikang [1]. The quality of drinking water in southern Tibet is not properly disinfected, which poses a substantial risk to drinking water safety [63].

The water quality in the Zhaxikang catchment ranges from excellent to extremely poor, with the EWQI value being closely related to high TDS, sulfate, nitrate, and B values (see Supplementary Tables S2 and S3). The average regional EWQI was 315.9, with values ranging from 14.8 to 1373.6. (Figure 11). The spatial distribution of water quality in Zhaxikang is a large variation. It results in geothermal water with very poor water quality being discharged into the river. The majority of the samples (42.9%) were classified as excellent, primarily upstream river water and shallow groundwater, while extremely poor samples accounted for 38.1% (eight samples), primarily hot spring water, polluted river, and part of cold spring. Two samples (9.5%), one sample (4.8%), and one sample (4.8%) were classified as poor, medium, and good quality, respectively, mostly mixed water and shallow groundwater.

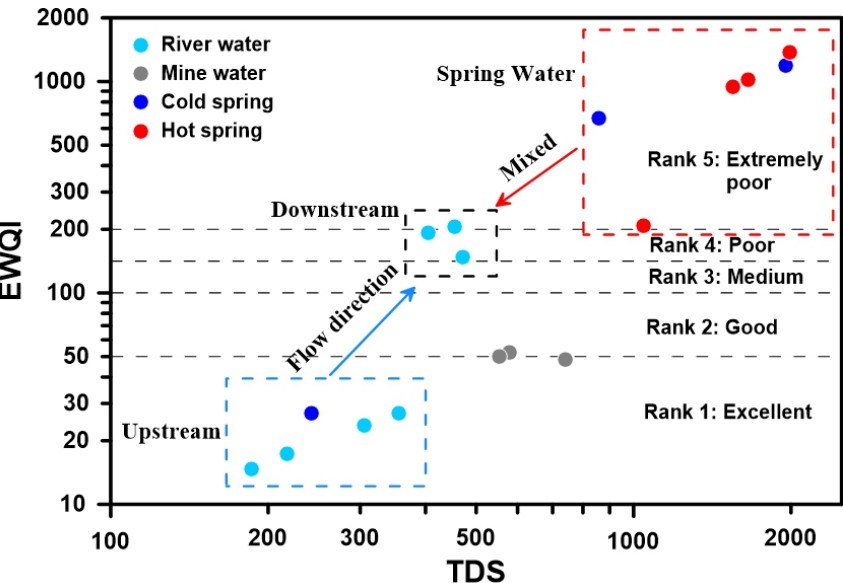

**Figure 11.** Correlation diagram between EWQI and TDS.

According to the results of EWQI (Figure 11), there are two end elements of water quality in the Zhaxikang area, drinkable water and polluted water, and the mixing of them is the primary cause of the deterioration of water quality in the Zhaxikang area.

Dilution is the main self-purification process of rivers. In addition, the concentration of B in the river downstream from the geothermal water discharge increased slightly from R01 to R02 and then decreased to R03 (Figure 10b). The increase in B concentration at R02 could be explained by the fact that the point is located near a fault, and there could be hidden hot spring seepage at the river's bed. According to Figure 10b, B concentration is significantly reduced due to the action of self-purification of river water; however, its value is still far above the limits of drinking water.

After the geothermal water is discharged into the river, all three mixed samples (R01, R02, and R03) have a HI (Hazard Index) value higher than 1, indicating that the noncarcinogenic risk is unacceptable [24,64]. The noncarcinogenic risk factor for children ranged from 0.002 to 1.844, with an average of 0.375, whereas the HQ (Hazard Quotient)

for adults ranged from 0.002 to 1.580 (see Supplementary Table S5). Generally, children are at a higher health risk than adults [18,65]. Based on the perspective of spatial analysis of geothermal water pollution, the $HQ_{Ingestion}$ and $HQ_{Dermal}$ for the downstream water samples (R01 and later) after geothermal water mixing are both higher than 1, indicating that geothermal water mixing is the primary source of drinking water safety hazards in the Zhaxikang area.

*3.5. Contamination Model of Drinking Water in Zhaxikang Area*

Geothermal water is a potentially serious source of contamination, and water quality will be significantly affected as geothermal water is discharged into the river directly. Water quality is affected by a variety of factors, including topography, bedrock properties, soil characteristics, and meteoric precipitation [66].

This study identified the main water pollution sources, mixing patterns, and health risks in the Zhaxikang area using hydrochemistry, isotope analysis, and risk assessment. Figure 12 shows the mixed contamination model of geothermal water to surface water and shallow groundwater in the Zhaxikang area. Rainwater and snow meltwater are the main recharge sources of cold groundwater in this area, while thermal groundwater also has a magmatic water source. The deep hot water will mix with the shallow groundwater along the faults during the upward migration process, changing the physical and chemical properties of the shallow groundwater [37]. Additionally, the existence of deposit veins will also affect the surrounding groundwater properties [33].

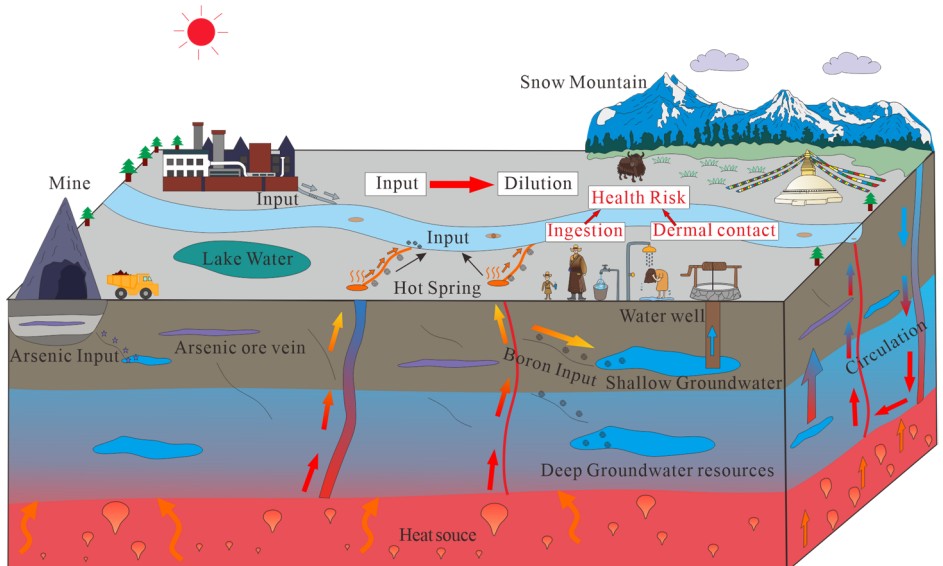

**Figure 12.** Schematic contamination model of geothermal water to surface water and shallow groundwater in Zhaxikang Area.

Given the drinking water situation in Tibet, the presence of pollution sources in river water may pose some health risks to nearby residents' drinking water sources, including drinking water safety hazards and skin contact hazards. Geothermal water is drained into the river with excessive ion concentration (B, $HCO_3^-$, As) in the form of hot springs and seepage on the riverbed. Relatively speaking, the seepage in the river is a relatively hidden pollution source, which is not easy to detect. On the other hand, mining would also release arsenic, which can also contaminate shallow groundwater and surface water [33]. Arsenic concentrations in lake water and mine water, for example, exceed limits in the study area. The residents of the area are at risk from the high boron water produced by the mixing of geothermal water when these shallow groundwater and river waters are used for drinking water.

*3.6. Prospect of Water Security in Geothermal Development*

The Qinghai–Tibet Plateau region has the most abundant and concentrated geothermal resources in China [67–70]. Geothermal exploitation and utilization in the study area will enter a new era under the carbon-neutral framework. The pollution of geothermal water to surface water will become a major issue of security for this region's water resources.

The Zhaxikang thermal field has not been developed and utilized on a large scale, and the harmful elements mainly come from the discharge of hot springs and mining wastewater. However, as the development and utilization of geothermal energy accelerate, the discharge and treatment of geothermal wastewater must be addressed in the future. For example, there were no endemic diseases in the Yangbajing area during geothermal exploration, but the residents experienced an increase in dental fluorosis, baldness, and other diseases after geothermal wastewater was discharged into the Zangbuqu River without proper treatment [12]. The toxic and harmful elements carried by geothermal water cause the water quality of the river to exceed the drinking water standard, even after the self-purification effect of the river.

As a result, to protect Zhaxikang's water environment, the problem of geothermal wastewater discharge and purification must be clarified. Reinjecting geothermal wastewater into the reservoir is an environmentally friendly and effective way to avoid surface pollution and maintain the geothermal fluid pressure that is necessary for geothermal recovery. On the other hand, it is necessary to strengthen monitoring of the geothermal mining industries and other industries' pollutant output to the environment, as well as to impose appropriate restrictions and regulations. In addition, local governments and relevant organizations also need to educate the residents about public health, strengthen the protection of children, and enhance awareness and knowledge of drinking water treatment.

## 4. Conclusions

Based on the hydrochemical and isotopic characteristics of hot springs, cold springs, river water, cold groundwater, and lake water in the Zhaxikang catchment, this paper reveals the water–rock interaction mechanism controlling the water chemical characteristics and B pollution source, as well as evaluates the potential water environmental health risk of geothermal water discharge. The main conclusion could be as follows:

1.  The geothermal water is characterized by high boron content in the study area, and the chemical type of hot spring water is $HCO_3$-Cl-Na, which is related to the metamorphic $CO_2$ degassing and mixing process in the geothermal area.
2.  Hydrogen and oxygen isotope, Gibbs diagram, Pearson correlation, and ion correlation were used to explore the evolution mechanism of groundwater quality. Ice melting water, meteoric water, and magmatic water are all sources of geothermal water. The groundwater is an open system, and the major components are controlled by water–rock processes such as silicate mineral dissolution and cation exchange.
3.  B mainly comes from deep volatiles, cation exchange, and leaching in shallow sediments.
4.  According to the EWQI results, the water quality of the river gradually deteriorated after flowing through the geothermal area and mixed with geothermal water (with a mix ratio of ~20%). Additionally, there are significant non-carcinogenic health risks if drinking contaminated water, and children are at a higher health risk than adults.

In the context of energy conservation and emission reduction, the exploitation of the rich geothermal resources in the Qinghai–Tibet Plateau will be accelerated. Many toxic and harmful elements of geothermal water discharged into the river will be a threat to the regional water environment and health risks. Therefore, attention should be paid to environmental monitoring during geothermal exploitation and its impact on the water environment and the drinking water safety of residents.

**Supplementary Materials:** The following supporting information can be downloaded at: https://www.mdpi.com/article/10.3390/w14203243/s1, Figure S1: Relationship of the B concentration with other measured hydrochemical parameters; Table S1: Information entropy and entropy weight of parameters; Table S2: Assessment results according to the calculated EWQI; Table S3: Water quality class based on EWQI; Table S4: Model Parameters for the calculation of exposure dose; Table S5: Assessment results of health risks based on drinking water intake and dermal contact. References [71–84] are cited in the Supplementary Materials.

**Author Contributions:** Y.W. provided supervision in the presented work and involved in the writing and editing of the manuscript. L.L. (Liang Li) conducted the investigation and data collection, developed the suitable methodology, and was involved with the writing of the manuscript. H.G., L.L. (Lianghua Lu) and L.L. (Luping Li) helped in methodology selection and visualization, and participated in editing of the earlier versions of the manuscript. J.P. and F.C. helped in figure preparation and edited and wrote the early version of the manuscript. All authors have read and agreed to the published version of the manuscript.

**Funding:** This study was supported by the National Natural Science Foundation of China (42002300, 41907174), the China Geological Survey's project: DD20211381, the National Key Research and Development Program of China (2018YFC0406506), and the Open Fund (PLC2020032) of the State Key Laboratory of Oil and Gas Reservoir Geology and Exploitation (Chengdu University of Technology).

**Institutional Review Board Statement:** No applicable.

**Informed Consent Statement:** No applicable.

**Data Availability Statement:** Not applicable.

**Acknowledgments:** We are grateful to the editors and the three anonymous reviewers. Their comments are quite constructive and helpful for us to improve the quality of the manuscript.

**Conflicts of Interest:** The authors declare no conflict of interest.

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
