# Peer review of "The Genesis Mechanism and Health Risk Assessment of High Boron Water in the Zhaxikang Geothermal Area, South Tibet"

_water, doi:10.3390/w14203243_

Round 1

Reviewer 1 Report

I have read the manuscript entitled: “

 The Genesis Mechanism and Health Risk Assessment of High 2 Boron Water in the Zhaxikang Geothermal Area, South Tibet” and I would like to address following suggestions to the authors:

1.      The abstract must contain a presentation of the manuscript with the main results obtained, please rewrite the abstract.

2.      Please remove highlights from the manuscript.

3.      Introduction part must be improved and remove - 2. Hydrogeological Settings”.   

4.      Please insert Hydrogeological Settings in section Materials and Methods that be written as study area. Please write metallic cations as: Fe3+, Cu2+, Pb2+, Cr3+, Ni2+, Cd2+, and As5+ if have this form. This information is supplied on CRM bottles used.

5.      Please insert superscript and subscript at unit of measure and chemical formula for all metallic ions and reagent used in the manuscript.  

6.      Please rewrite the conclusions for each study presented as a result, a conclusion must be presented.

Reviewer 2 Report

Dear authors,

I enjoyed your manuscript and water data analysis of the Zhaxikang Geothermal Area. I've only a few minor comments and suggestions you can find in the attached commented pdf file. In data discussion, pay attention to repeating the same concepts - it's not a relevant issue but makes the reading less fluent, thus I suggest you try to fix this with your revisions; this will further improve the quality of this paper.

Best regards 

Author Response

Response to Reviewer 2 Comments

Dear reviewer, thank you for revising the grammar and text of our manuscript. We have corrected the manuscript content, figures, and tables according to the comments in the pdf. Thank you again for your recognition of our manuscript. In response to your valuable comments, we have made corresponding adjustments to the manuscript, and the specific revisions are as follows:

Point 1. The texts in this figure are somewhat hard to read, especially the yellow texts in fig. a, or names of sampling points in fig. c.

Response 1: Dear reviewer, thank you for your comment, we have modified the figure 1 as you suggested.

Point 2. This sentence is unclear - do you mean risk assessment for  non carcinogenic effects associated to B oral exposure was performed according to USEPA (2008)?

Response 2: Dear reviewer, we are very sorry for the misunderstanding caused by our misrepresentation. In this study, we used the empirical model proposed by the USEPA to assess the health risks of high boron geothermal water, We changed the text to ” To assess the health risk of ground and surface water mixed with geothermal water, this study quantified health risks from drinking water intake and dermal contact using the empirical model proposed by the United States Environmental Protection Agency (USEPA, 2008)”

Point 3. Even the dataset is limited (n=3 samples for certain water types), I suggest adding a figure (histograms or box plots) to summarize the data from table 2, and help comparison among different waters types.

Response 3: Thank you for your valuable suggestions, we have added a box plots figure to summarize the data from Table 2.

Point 4. But S02 it's cold spring water, doesn't it?

Response 4: Dear Reviewer, yes, you are right. S05 and S02 are both spring water samples, which represent groundwater. We have modified "other hot spring water samples" to "other spring water samples". Thank you for correcting the text of the manuscript.

Reviewer 3 Report

The article ,,The Genesis Mechanism and Health Risk Assessment of High Boron Water in the Zhaxikang Geothermal Area, South Tibethas’’ has theoretical and practical value. The authors found that downstream health risks are high due to the high mixing ratio of geothermal water. There are no errors in the article. It is proposed to fix the layout of Table 2.

Author Response

Response to Reviewer 2 Comments

Point 1. The article “The Genesis Mechanism and Health Risk Assessment of High Boron Water in the Zhaxikang Geothermal Area, South Tibethas’’ has theoretical and practical value. The authors found that downstream health risks are high due to the high mixing ratio of geothermal water. There are no errors in the article. It is proposed to fix the layout of Table 2.

Response 1: Dear reviewer, on behalf of my co-authors, we are honored that this manuscript can be recognized by you. Your revisions have greatly improved the manuscript and made the manuscript read more smoothly. We have revised the manuscript accordingly based on your comments. Your rigorous academic attitude is an example for us to learn from, thank you. Please refer to the attachment for the revised manuscript.
